# Dead but Not Dead End: Multifunctional Role of Dead Organs Enclosing Embryos in Seed Biology

**DOI:** 10.3390/ijms21218024

**Published:** 2020-10-28

**Authors:** Gideon Grafi

**Affiliations:** French Associates Institute for Agriculture and Biotechnology of Drylands, Jacob Blaustein Institutes for Desert Research, Ben-Gurion University of the Negev, Midreshet Ben Gurion 84990, Israel; ggrafi@bgu.ac.il

**Keywords:** dispersal unit, seeds, fruits, florets, spikelets, seed coat, pericarp, glumes, husk, dead organs enclosing embryos, storage organs, hydrolases, nutrients, phytohormones

## Abstract

Dry fruits consist of two types, dehiscent and indehiscent, whereby the fruit is splitting open or remains closed at maturity, respectively. The seed, the dispersal unit (DU) of dehiscent fruits, is composed of three major parts, the embryo and the food reserve, encapsulated by the maternally-derived organ, the seed coat. Indehiscent fruit constitutes the DU in which the embryo is covered by two protective layers (PLs), the seed coat and the fruit coat. In grasses, the caryopsis, a one-seeded fruit, can be further enclosed by the floral bracts to generate two types of DUs, florets and spikelets. All protective layers enclosing the embryo undergo programmed cell death (PCD) at maturation and are thought to provide mainly a physical shield for embryo protection and a means for dispersal. In this review article, I wish to highlight the elaborate function of these dead organs enclosing the embryo as unique storage structures for beneficial substances and discuss their potential role in seed biology and ecology.

## 1. Introduction

The seed is the fundamental unit of dispersal in higher plants where the embryo is encapsulated by maternally-derived dead coverings until the seed germinates and resumes growth. Yet plants have evolved a variety of dispersal units (DUs) including fruits, florets, and spikelets. Dry fruits consist of two major groups, dehiscent, in which the fruit is splitting open at maturity to allow for seed dispersal, and indehiscent, whereby the fruit is not open at maturity and constitutes the DU (Figure 1). Thus, the DU may be categorized based on the number of dead organs enclosing the embryo (DOEEs) [1]. Accordingly, the seed is the DU of dry dehiscent fruits (e.g., *Arabidopsis thaliana*), whereby the embryo is covered by a single layer, the seed coat (Figure 1A). The fruit represents the DU of dry indehiscent fruit (e.g., *Medicago* spp.) and therefore the embryo is covered by two layers, the seed coat and the fruit coat (the pericarp and its accessories) (Figure 1B). Notably, caryopsis (e.g., Poaceae species) and achene (e.g., Asteraceae species) are types of dry one-seeded indehiscent fruits whereby the seed coat and the pericarp are either fused or separated, respectively. In grasses, the basic DU, the caryopsis is often covered by the hardened floral bracts to generate two basic types of DUs, the floret and the spikelet (Figure 1C).

The seed is essentially composed of an embryo, a food reserve (endosperm, cotyledons, hypocotyl, perisperm) and various numbers of dead layers (seed coat, pericarps, floral bracts) commonly thought to provide a physical shield for embryo protection and a means for dispersal [2,3]. The food reserve is essentially of zygotic origin and is often made up of embryonic organs (cotyledons and hypocotyl) except for the perisperm (perispermous seeds e.g., black pepper) where it is of maternal origin and derived from the nucellus or the integuments. In this review article, I will elaborate on commonly-ignored, maternally-derived organs, namely DOEEs, as integral parts of the dispersal unit (DU) and highlight their function as a unique storage entity providing nutrition as well as regulatory substances that could affect seed biology and ecology.

## 2. The Biological Significance of DOEEs

All protective layers enclosing the embryo, namely, the seed coat, the pericarp, lemmas, paleae, and glumes are maternally-derived and are dead at maturity. Although the term dispersal unit highlights an entity that is specialized for dispersal, published studies have shown that the dead, maternal part of the dispersal unit serves multiple functions including protection from predation, seed positioning in the soil, moisture adsorption, seed anchoring, light filtering, regulation of seed respiration, dormancy, and germination [4,5,6,7,8,9,10,11,12,13,14]. Several studies pointed to nutrients such as potassium, calcium, and other cations that are stored in the bracts of wild wheat and *Eurotia* and move into the seed during imbibition [15,16]. Some work demonstrated that intact dispersal units improve the seedling establishment compared to naked seeds [16,17,18,19]. In a recent study, El-Keblawy et al. [20] examined the importance of the husks of *Brachypodium hybridum* DU to germination and seedling growth. They found that the husks, particularly soaked husks, significantly improved all examined parameters including final germination and the germination rate index as well as seedling growth. Similarly, germination from the intact DUs of *Avena sterilis* and wild emmer wheat is significantly improved compared to naked caryopses at all parameters examined including final germination percentage and germination rate [19]. However, the effect of husks on germination appears to be species-specific. Accordingly, earlier work on rice showed that the rice husks contain substances that play a significant role in inhibiting seed germination [21]. On the other hand, Ueno and Miyoshi [22] reported that germination of intact rice seeds collected at 45 days after anthesis was significantly higher than those of de-husked seeds at temperatures ranging from 20 to 34 °C.

## 3. DOEEs Function as Long-Term Storage for Beneficial Proteins: Effect of Maternal Environment

At maturation, all organs enclosing the embryo, including seed coat, pericarp, and floral bracts, undergo developmentally induced programmed cell death (PCD) [23,24,25]. PCD is a physiological process that controls the disassembly and removal of cell components (except for cell walls). Two major types of PCD have been described, namely environmental and developmental PCD [26]. PCD involves the degradation of almost all macromolecules such as DNA, RNA, and proteins whose constituents are remobilized into other plant parts [27,28,29]. Recent reports, however, have demonstrated a type of PCD occurring in seed coverings where hundreds of proteins remain intact and can persist in active form for decades, and are released to the immediate surrounding of the dispersal unit upon hydration [16,19,30,31].

Proteome analysis performed on proteins extracted from DOEEs of various plant species revealed hundreds of proteins, which were categorized into multiple molecular function groups including catalytic activity, binding activity, hydrolase, pectinesterase, and oxireductase activities. Classification for biological processes showed that many of the released proteins are involved in metabolic processes and oxidation–reduction processes as well as in response to stimulus [1,19,30]. Within the group of proteins responsive to biotic and abiotic stimulus are several defensin-like (DEFL) proteins involved in response to pathogenic fungi [32]. Indeed, a higher level (24-fold) of defensin was released from *Anastatica hierochuntica* seeds derived from plants subjected to salt stress compared to seeds from non-stressed plants [30]. Other proteins implicated in conferring resistance to pathogenic fungi include chitinases and gluconases [33,34,35]. The proteomic data also listed S1 type endonucleases in seed coats and pericarps of *A. hierochuntica* [1,19,30] whose activity was reduced following exposure to salt or to a combination of salt and heat [30]. On the other hand, proteases commonly accumulated in DOEEs showed a significant increase in activity in pericarps derived from *Anastatica* plants treated with salt and heat [30]. Endonuclease as well as protease activities were enhanced in husks derived from *A. sterilis* plants experiencing drought [19]. Endonucleases, in general, have been implicated in diverse cellular processes including fragmentation of genomic DNA during PCD [36,37,38] and RNases were shown to inhibit growth of various pathogenic fungi [39,40]. Intriguingly, hydrolytic enzymes in DOEEs can retain their integrity and activity even 50 years after the death of the tissue [31].

Multiple proteins involved in reactive oxygen species (ROS) detoxification are accumulated in DOEEs [1,19,30]. These include superoxide dismutases (SODs) that catalyze the conversion of oxygen radicals such as superoxide into hydrogen peroxide (H_2_O_2_). Catalases and ascorbate peroxidases (APXs) are responsible for H_2_O_2_ detoxification [41,42] and glutathione peroxidases catalyze the reduction of H_2_O_2_ and hydroperoxides to H_2_O or alcohols [43]. Some of the ROS detoxifying enzymes such as SOD, catalase, and peroxidase were accumulated to a higher level in husks of *Avena sterilis* subjected to drought conditions or in pericarps of salt-treated *A. hierochuntica* plants [19,30]. ROS appear to carry out important roles in seed biology and physiology [44,45]. Although, ROS are destructive elements of live tissues they can act as signaling molecules that play a role in seed longevity, seed dormancy release, seed germination, and in defense against potential soil pathogens [44,45,46].

Proteomic data showed the accumulation of cell-wall-modifying enzymes such as pectinesterases (PMEs/PEs), pectin lyases, and polygalacturonases (PGs) in DOEEs [1,19,30]. These enzymes break down pectin, the major constituent of plant cell walls and are involved in multiple developmental processes in plants. PMEs and PGs were found in germinating seeds of various plant species and are involved in softening cell walls in preparation for radicle emergence [47,48,49]. Thus, in addition to a zygotic source of PMEs and PGs, DOEEs provide a complementary, maternally source for cell-wall-modifying enzymes to ensure proper radicle protrusion and seed germination. Notably, among the cell-wall-modifying enzymes accumulated in DOEEs, several pectinesterase/pectinesterase inhibitor, also known as pectin methylesterase inhibitor (PMEI) enzymes were up-accumulated in pericarps of salt-treated *A. hierochuntica* plants [30]. These enzymes were shown to negate PME activities and exhibit antifungal activity against various pathogens including *Fusarium oxysporum* f. sp. *matthiole*, *Alternaria brassicicola*, *Botrytis cinerea*, and *Verticillium* wilt [50,51].

## 4. Heat Shock Proteins (HSPs)

The accumulation of HSPs (e.g., HSP90, HSP70, and sHSPs) in DOEEs appears to be a general phenomenon found in all species examined, with a notable effect of maternal environment on their accumulated levels [19,30,31]. Generally, HSPs are molecular chaperones encoded by a multigene family and are involved in diverse cellular processes including folding of newly-synthesized proteins and transport of precursors within the cell as well as in targeting unfolded or damaged proteins for degradation. The HSPs of the HSP70 class are most abundant in the dead pericarps of *A. hierochuntica* or released from the seeds [30]. Small HSPs (sHSPs) are notable in the husk of *Avena sterilis* [19] but also found in pericarps and seeds of *A. hierochuntica* [30]. The sHSP family is highly diverse in angiosperms and consists of various types of sHSPs that can be found in the cytosol or targeted to various organelles including the nucleus, ER, chloroplasts, mitochondria, and peroxisome [52]. Notably, under normal, non-stressed conditions sHSPs expression is developmentally regulated and often restricted to certain developmental stages including embryo and pollen development and germination [53]. However, sHSPs are commonly induced following exposure to various stress conditions including heat, drought, and salt and have been correlated with stress tolerance [53]. Indeed, the sHSP16.9 was significantly up-accumulated (~25-fold) in husks derived from *A. sterilis* plants experiencing drought [19]. The mechanism by which sHSPs confer stress tolerance is not well understood. Some sHSPs were shown to act as molecular chaperones stabilizing and assisting proper protein folding [54,55]. It is thus possible that HSPs accumulated in DOEEs are physically associated with multiple proteins, perhaps to avoid their unfolding and aggregation, maintaining them in a competent state for correct folding [55] that enables them to promptly resume activity upon rehydration and seed germination.

## 5. DOEEs Contain Plant Growth-Promoting Activity

Previous reports have demonstrated the beneficial effect of DOEEs on seed germination and seedling establishment. Accordingly, seedlings of emmer wheat germinated on a germination paper from the intact DU performed better than those germinating from the naked caryopsis showing significant increase in number and length of lateral roots [16]. Also, germination assays performed on red sandy soil of winter wild oat and wild emmer wheat showed faster, greater, and more uniform germination from the intact DUs or the intact florets than germination from naked caryopses [19]. Likewise, hydrated husked grains of *Brachypodium hybridum* displayed enhanced germination under optimal light and temperature conditions compared to de-husked grains [20] and the hairy bracts of winterfat (*Eurotia lanata*) dispersal unit were shown to contribute profoundly to seedling establishment and vigor [17]. Presently, the mechanism(s) by which the husks promote and synchronize germination is not clear. Some of the mechanisms may be related to mechanical effects where husks may serve as a check point controlling radicle protrusion or the inflow/outflow of water and gases as well as being a source for multiple substances (proteins and metabolites) that promote or inhibit germination.

Some of the effects of DOEEs on seed performance and fate might be attributed to their function as rich storage for nutrients such as potassium, calcium, phosphorus, and sulfur thus providing an immediate nutritional supply for germinating seeds [1,15,56,57,58]. The beneficial effect of DOEEs on germination and seedling performance could be mediated by phytohormones that are released upon hydration either inside toward the embryo or outside to the immediate environment of the seed. Indeed, metabolic analysis of glumes of wild emmer wheat DU and pericarps of *Anastatica hierochuntica*, as well as husks of *Avena sterilis* revealed the presence of multiple phytohormones and plant growth factors including ABA, IAA, cytokinins, jasmonic acid (JA), and salicylic acid (SA) [19,30]. Both JA and SA play a central role in plant immunity [59] and might be involved in priming the germinating seeds in preparation to biotic stresses [60,61]. Notably, ABA is well known for its effect on seed dormancy [62], yet the level of ABA stored in the husk may not be sufficient for inhibition of germination but rather for stress-response signaling [62,63]. Alternatively, the presence of cytokinins such as trans-Zeatin (tZ) and dihydrozeatin (DHZ) in DOEEs [30] may antagonize the inhibitory effect of ABA on germination [64,65]. Although ABA appears to antagonize the activity of JA and SA in inducing disease resistance, it is possible that the combined action of these stress regulators at specific concentration levels is complementary in priming the germinating seeds for both biotic and abiotic stresses [60,66].

## 6. DOEEs Accumulate Allelopathic Substances That Affect Germination of Neighboring Seeds

Seeds and pericarps of *A. hierochuntica* release, upon hydration, allelopathic substances (germination inhibitors) that act in a species-specific manner to inhibit seed germination of certain plant species. Thus, seeds and pericarps possess substances that strongly inhibited germination of *Sinapis alba* (white mustard) seeds but had no or slight effect on seed germination of *Brassica juncea* and *A. hierochuntica.* Likewise, the husks of *Avena* DUs release, upon hydration, allelopathic substances that inhibit seed germination of *S. alba*, but have no notable effect on seed germination of *B. juncea.* Selective allelopathy described previously [67,68] may have an adaptive value as it can reduce competition for resources by neighboring plant species [69], but at the same time permitting seed germination of other species to maintain facilitative plant–plant interaction [70].

## 7. DOEEs and Microbial Growth

DOEEs store and release upon hydration, substances that can either promote or inhibit microbial growth depending on the plant species. While extracts from *A. sterilis* caryopses had no effect on microbial growth, extracts of *Avena* husk promoted microbial growth [19]. Similarly, substances released from *S. alba* seeds and pericarps promoted microbial growth with pericarp extracts displaying the highest promotive effect [58]. On the other hand, a strong inhibitory effect on microbial growth was exerted by substances released from *A. hierochuntica* seed coat and pericarps [30,31]. Inhibitory effect on growth has been reported for grains of a *Sorghum* cultivar that inhibit growth and fermentation of *Lactobacillus leichmannii* and *Saccharomyces cerevisiae*; the inhibitory factor is located in the pericarp fraction [71]. Similarly, seeds of *Abutilon theophrasti* Medik released a water-diffusible substance(s) that inhibited the growth of many types of bacteria and fungi [72]. The differential effect of DOEEs on microbial growth is probably related to the mode of interaction between plants and their unique habitat and the co-evolution with their specific microbiota. We assume that *Avena* husks release substances that promote microbial growth, which in turn produce and release substances such as plant growth regulators and defense inducers [73] that facilitate survival and growth and development. On the other hand, strong inhibition of bacterial growth might be necessary to combat potential soil pathogens that are prevalent in the ecosystem.

## 8. The Effect of Climate Change on DOEE Properties

Abiotic stresses are expected to increase in frequency and severity as a result of global climate change affecting large areas around the world [74]. The Mediterranean basin is particularly experiencing an increase in severity of droughts due to continuous reduction in precipitation and shortening of the wet season, which is accompanied by a prolonged, high temperature summer [75,76,77]. Most work addressing plant behavior under climate change focused on phenological and physiological processes that eventually affect plant population dynamics and diversity. However, we should be aware that the most important factor in determining dynamics and diversity of plants is the dispersal unit [78] whose properties (particularly the maternal component of the DU, that is the DOEE) have not been sufficiently studied under climate-change-induced adverse environmental conditions.

Obviously, DOEEs appear to be a central component of plant reproduction and seed biology and ecology whose physical, chemical, and biochemical properties are affected by maternal environment. Earlier work focused mainly on physical properties of the seed coat demonstrating that soybean grown in liquid solution with reduced concentrations of minerals and cytokinin developed seeds with thicker, less permeable coats that were poorly germinated [79]. Similarly, water stress imposed on soybeans resulted in increased water impermeability of seeds and in low percentages of ruptured seed coats [80]. Achenes of *Chenopodium bonus-henricus* collected from high elevation had thicker seed coats containing increased levels of polyphenols [81]. Day-length at the time of seed development and maturation of *Ononis sicula* and *Trigonella arabica* (both Fabaceae species) was shown to affect seed coat properties including color and permeability [82]. Recent studies have analyzed the effect of maternal environment on DOEE chemical and biochemical properties showing major effects on the composition and level of proteins and other substances accumulated within DOEEs [19,30]. Thus, *Avena sterilis* plants experiencing drought conditions (naturally low precipitation of 155 mm or simulated drought from 675 mm to 180 mm via rainout shelters) displayed changes in composition and levels of multiple proteins accumulated in the husks [19]. These included an increase in chitin-binding type 1 protein (~40-fold), involved in defense against pathogenic fungi [83]; sHSP16.9 (~25-fold), involved in stress tolerance; and the copper/zinc superoxide dismutase chaperone (~18-fold), involved in ROS detoxification [84]. Furthermore, husk derived from plants grown under drought conditions showed a significant increase in ABA and certain amino acids including proline, serine, alanine, and leucine as well as the non-protein amino acid γ-aminobutyric acid (GABA). Overproduction of proline is a well-known phenomenon in plants responding to biotic and abiotic stresses, imparting stress tolerance at least partly by maintaining the osmotic balance of the cell and by mitigating oxidative burst [85]. Sugars (e.g., glucose and fructose) were accumulated to high levels in *Avena* husks derived from plants experiencing drought [19] as well as in *A. hierochuntica* pericarps derived from salt-treated plants [30]. Further changes in pericarp properties were found in *A. hierochuntica* following exposure to salt. Accordingly, the levels of IAA and ABA were reduced significantly while those of salicylic acid and the cytokinin tZ were significantly increased in dead pericarps of stressed plants. Most notable is a major reduction in the intermediates of the tricarboxylic acid (TCA) cycle following salt treatment including citrate, succinic acid, malic acid, and fumaric acid [30], which might reflect their abundance in the live pericarp tissues in response to salt. Indeed, plants subjected to salinity undergo physiological stress leading to reduction in abundance of the TCA cycle intermediates, concomitantly with increased levels of certain amino acids including proline [86].

## 9. Concluding Remarks

DOEEs are emerging as a major multifunctional component of the dispersal unit that have been uniquely evolved in each plant species in conjunction with its unique habitat to nurture the embryo and to ensure its success in the habitat. Besides providing a physical shield for embryo protection and a means for dispersal, DOEEs function as storage structures for nutrition, proteins, and other regulatory substances (Figure 2) that act together as an organized unit to ensure the survival of the embryo during seed storage in the soil, and its proper growth and establishment in the habitat.

Stress conditions that are expected to increase in severity, over large areas of the world, due to climate change not only modify the embryo properties but also the properties of the DOEEs, which in turn impact seed performance and fate, soil properties, and consequently plant population dynamics and diversity [78,87,88].

Considering the role of DOEEs as a natural coating and their importance for the performance and fate of the embryo, we should reexamine all kinds of practices operating in agriculture as well as in gene banks where DOEEs are often removed (agricultural waste), and naked seeds are further processed for various applications. Accordingly, a common practice in agriculture for enhancing seed performance is through the addition (coating) of hazardous chemicals to protect the germinating seed from pest and pathogens, as well as adding plant growth promoting microorganisms and other substances for increasing germination success. One major problem in seed coating is the prophylactic nature of the coating where chemicals are added even if they are not strictly necessary, causing land and surrounding areas to become extremely toxic to pollinators, animals, and humans [89,90]. We should examine sowing seeds with their natural coating when it is appropriate or use an alternative environmental-friendly coating with substances derived from DOEEs or from other agricultural waste to increase germination rate, bacterial/fungal pathogens resistance, and seedling performance [91]. Indeed, recent work demonstrated the efficiency of coating seeds with plant-based proteins and other substances to enhance early plant growth and development [92,93].

Finally, seed bank protocols usually include a cleaning stage whereby seeds or caryopses are separated from their natural coating that constitutes the dispersal unit to avoid contamination by potential pathogens and to save storage space [94]. However, the emerging roles of seed coverings in storage of beneficial substances should trigger discussion regarding the most efficient way of storing plant genetic resources. Indeed, Rao et al. [94], addressing seed quality in gene banks, described various agronomic approaches with potential to improve quality and longevity of stored seeds in seed banks. In this review, a few examples from past studies are given, which imply a possible role of seed coverings (glumes, shells etc.) in maintaining seed longevity. Accordingly, several aspects of ex situ conservation may be considered and addressed in future work:Does preservation of the whole dispersal unit has an advantage over naked seeds for longevity, germination and seedling establishment?How do maternal growth conditions affect DU properties?Can we develop markers for the assessment of seed viability based on the activity of enzymes or metabolites stored within DOEEs?

Such research can be carried out promptly and effectively through collaborative studies with gene banks and herbaria that possess seeds and dispersal units, which have been stored under various conditions for various periods of time.

## Figures and Tables

**Figure 1 ijms-21-08024-f001:**
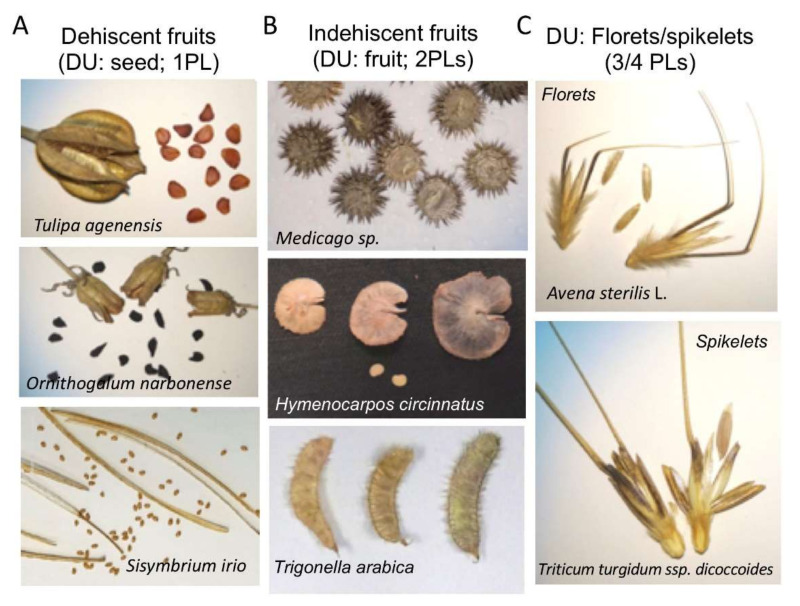
Examples of various dispersal units (DUs) in angiosperms. (**A**) Dehiscent fruits, the seed represents the DU and embryo in covered by a single protective layer (1PL). (**B**) Indehiscent fruits, which constitute the DUs and the embryo is covered by 2PLs. (**C**) Common DUs in grasses whereby the floral bracts (glumes, lemmas, paleae) enclose the caryopsis (a one seeded fruit), to form two types of DUs, florets and spikelets, and thus the embryo is covered by three and four PLs, respectively.

**Figure 2 ijms-21-08024-f002:**
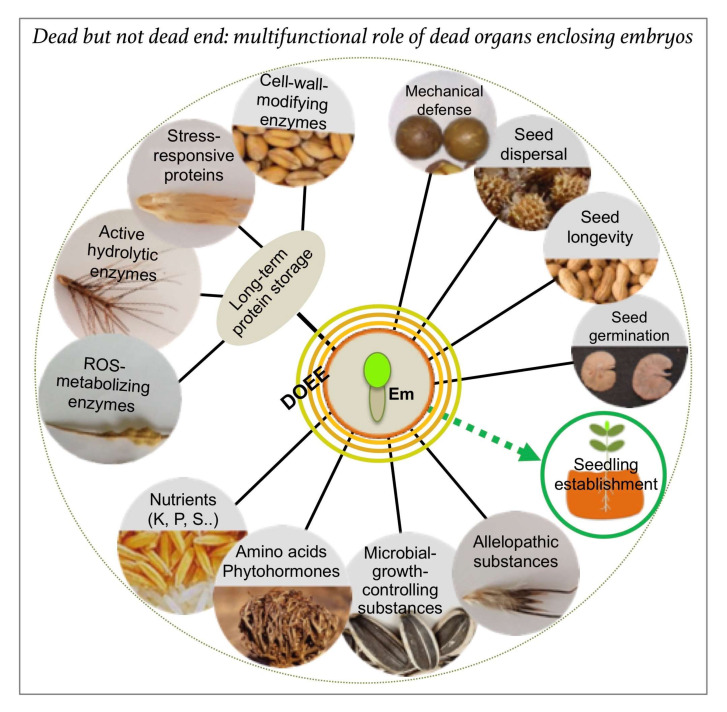
Dead organs enclosing the embryo (DOEEs) represent multifunctional structures that provide mechanical defense and accessories for dispersal together with a rich storage for nutrients, proteins, and other regulatory substances. The embryo (Em) can be covered with one to four dead layers (colored circles) that act together to ensure offspring success in the habitat providing multiple beneficial proteins whose activities can persist for decades within the dead organs. Other substances, including microbial growth and allelopathic substances, nutrients, amino acids and phytohormones, may control longevity, growth, and development to bring about seedling vigor and establishment.

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
