# Peer review of "Dead but Not Dead End: Multifunctional Role of Dead Organs Enclosing Embryos in Seed Biology"

_ijms, 2020, doi:10.3390/ijms21218024_

Round 1

Reviewer 1 Report

A remarkable review paper on seed biology and novel in aspect of highlighting the natural materials of testa of each crop and how these dead materials can be helpful on germination and seedling establishment and even useful in modern seed technology for seed encrustments and coating. I only have minor editorial comments and some suggestions to enhance the quality of already excellent well-written manuscript. 

Line 2-3: Please use upper case for highlighted letters.

Line 14: seed coat or testa which term is more common worldwide. Or maybe use "seed coat (testa)"

Line 15: protective layers (PLs),

Line 36: A             B                 C  (please add labels on top of each vertical collage of pictures)

Line 38: Figure 1. Also please add the figure labels as you are referring to Fig. 1B (line 46)

Line 50: maybe (Fig. 1C)

Line 71: [16-19]

Line 106 and 108: Suggestion: combination of salt and heat

Line 114: reactive oxygen species (ROS)

Line 147: P in HSP stands for proteins maybe you can replace it with "HSPs of the ..." or "HS proteins of the ..."

Line 149: it should be HSP? not HPS

Line 168: blotter paper or standard roll towel paper?

Line 171: more uniform germination

Line 217: S. alba is Sinapis alba (white mustard)?

Line 291: Figure 2.

Line 316 and 317: in fact Amirkhani et al 2016, 2019 and and Qiu et al 2020, used the cellulose from newspapers natural-plant-based products as well as different sources of vermicompost and recorded germination rate increase, crop uniformity enhancement on different horticultural and cover crops such as broccoli, red clover and Italian ryegrass.

Amirkhani, M.; Mayton, H.S.; Netravali, A.N.; Taylor, A.G. A Seed Coating Delivery System for Bio-Based Biostimulants to Enhance Plant Growth. Sustainability 2019, 11, 5304. https://doi.org/10.3390/su11195304

Qiu, Y.; Amirkhani, M. (Co-first author); Mayton, H.; Chen, Z.; Taylor, A.G. Biostimulant Seed Coating Treatments to Improve Cover Crop Germination and Seedling Growth. Agronomy 2020, 10, 154. https://doi.org/10.3390/agronomy10020154

M. Amirkhani, Anil N. Netravali, Wencheng Huang, Alan G. Taylor (2016) Investigation of Soy Protein Based Biostimulant Seed Coating for Broccoli Seedling and Plant Growth Enhancement. HortScience 51(9):1121–1126. 2016. https://doi.org/10.21273/HORTSCI10913-16

Author Response

Thanks for your kind evaluation of the Review article and for the helpful comments that improved the Essay.

Reviewer 2 Report

This review article by G. Grafi (Dead but not dead end: multifunctional role of dead organs enclosing embryos in seed biology) summarizes the current knowledge about the role of the maternally-derived dead organs surrounding the embryos (so-called DOEE by the author) in protecting the seeds and enhancing seed germination.

The article covers a wide range of species and reviews many roles of these DOEE  such as germination promoting in normal and stressed conditions, anti-bacterial protection, nutrition and seed longevity.

I find the article well written and I appreciate very much the conclusion. This part may be more highlighted as a full paragraph than just the conclusion as it covers more than just synthesizing the parts written above.

I have two comments. 

Line 51-52, the author cites coty;edons and hypocotyl as food reserve. These 2 structures are part of the embryo. This should be clarified.

(There is a typo line 54, prisperm for perisperm.)

Line 87, there are two types of PCD, developmental and stress-induced. This should be clarified. The author may have a look at Van Durme and Nowack, COPB 2016 for clarification. 

Author Response

Thanks for kind evaluation of the Review article and for comments which I addressed in the attached file.
